# Molecular and Cellular Mechanisms of *M. tuberculosis* and SARS-CoV-2 Infections—Unexpected Similarities of Pathogenesis and What to Expect from Co-Infection

**DOI:** 10.3390/ijms23042235

**Published:** 2022-02-17

**Authors:** Anna A. Starshinova, Igor Kudryavtsev, Anna Malkova, Ulia Zinchenko, Vadim Karev, Dmitry Kudlay, Angela Glushkova, Anastasiya Y. Starshinova, Jose Dominguez, Raquel Villar-Hernández, Irina Dovgalyk, Piotr Yablonskiy

**Affiliations:** 1Almazov National Medical Research Centre, 197341 St. Petersburg, Russia; starshinova_aa@almazovcentre.ru (A.A.S.); vadimkarev@yandex.ru (V.K.); 2Institute of Experimental Medicine, 197376 St. Petersburg, Russia; igorek1981@yandex.ru; 3Laboratory of the Mosaic of Autoimmunity, St. Petersburg State University, 199034 St. Petersburg, Russia; piotr_yablonskii@mail.ru; 4St. Petersburg Research Institute of Phthisiopulmonology, 194064 St. Petersburg, Russia; ulia-zinchenko@yandex.ru (U.Z.); prdovgaluk@mail.ru (I.D.); 5Children’s Clinical Research Center for Infectious Diseases, 197022 St. Petersburg, Russia; 6Department of Pharmacology, Institute of Pharmacy, I.M. Sechenov First Moscow State Medical University, 119435 Moscow, Russia; d624254@gmail.com; 7Institute of Immunology, 115478 Moscow, Russia; 8V.M. Bekhterev National Research Medical Center for Psychiatry and Neurology, 192019 St. Petersburg, Russia; angela_glushkova@yahoo.com; 9Faculty of Medicine, St. Petersburg State Pediatric Medical University, 194100 St. Petersburg, Russia; asya.starshinova@mail.ru; 10Institut d’Investigació Germans Trias i Pujol, 08916 Barcelona, Spain; jadominguez@igtp.cat (J.D.); rvillar@igtp.cat (R.V.-H.); 11CIBER Enfermedades Respiratorias, 08029 Barcelona, Spain; 12Genetics and Microbiology Departament, Universitat Autònoma de Barcelona, 08193 Barcelona, Spain

**Keywords:** tuberculosis, SARS-CoV-2 infection, COVID-19, *M. tuberculosis* infection, Th cell subsets

## Abstract

Tuberculosis is still an important medical and social problem. In recent years, great strides have been made in the fight against *M. tuberculosis*, especially in the Russian Federation. However, the emergence of a new coronavirus infection (COVID-19) has led to the long-term isolation of the population on the one hand and to the relevance of using personal protective equipment on the other. Our knowledge regarding SARS-CoV-2-induced inflammation and tissue destruction is rapidly expanding, while our understanding of the pathology of human pulmonary tuberculosis gained through more the 100 years of research is still limited. This paper reviews the main molecular and cellular differences and similarities caused by *M. tuberculosis* and SARS-CoV-2 infections, as well as their critical immunological and pathomorphological features. Immune suppression caused by the SARS-CoV-2 virus may result in certain difficulties in the diagnosis and treatment of tuberculosis. Furthermore, long-term lymphopenia, hyperinflammation, lung tissue injury and imbalance in CD4+ T cell subsets associated with COVID-19 could propagate *M. tuberculosis* infection and disease progression.

## 1. Introduction

Tuberculosis is still an important medical and social problem in many countries of the world [1,2]. According to the WHO documents, tuberculosis remains one of the 10 diseases with the highest mortality rates in the world [3]. The number of new tuberculosis cases was estimated to increase from 8.8 million in 2010 to 10.4 million in 2016, and to decrease slightly to 10.0 million in 2018 (including 0.3 million cases with concomitant HIV) [2]. Tuberculosis mortality rates increased from 1.1 million in 2010 to a maximum of 1.5 million in 2013 and then decreased steadily to 1.3 million in 2020, when most tuberculosis cases were reported in South East Asia (44%) and South Africa (24%), and were three times less in the countries of the eastern Mediterranean (8%), European (3%) and American regions [4].

Drug-resistant tuberculosis has become the most urgent problem in recent years [3]. According to the WHO, 484,000 new cases of rifampicin-resistant tuberculosis were registered in 2018, with 78% of them being multiple drug-resistant tuberculosis (MDR-TB) [5]. In 2019, 50% of all MRD-TB and extensively drug-resistant tuberculosis (XDR-TB) cases were reported in India (27%), China (14%) and Eastern European countries (9%) [2]. The effectiveness of drug-resistant tuberculosis treatment does not exceed 56%. In 2018, 214,000 patients with XDR-TB died because they could not receive adequate therapy with new tuberculosis drugs [6].

The incidence of tuberculosis has gradually decreased among both adults and children in Russia since 2009 [7]. The official statistics indicate that it decreased by half (from 77.2 to 41.2 per 100,000), while the mortality decreased three times (from 15.4 to 5.2 per 100,000) from 2010 till 2019 [2]. However, the incidence of multidrug-resistant and extensively drug-resistant tuberculosis increased from 2010 to 2016 (22.0 to 25.8 per 100,000) and tended to decrease only since 2017 (24.7 to 21.4 per 100,000 in 2017 and 2019, respectively) [2,3]. Obviously, the administrative resource, improving the quality of screening, diagnosis and treatment of tuberculosis, makes a significant contribution to improving control over the spread of tuberculosis and reducing the incidence and mortality from the disease. In December 2019, the entire world community faced a new infectious agent, a novel coronavirus (SARS-CoV-2), which causes COVID-19 [8]. The first cases of a novel coronavirus infection were reported in Wuhan, Hubei Province, China at the end of December 2019 [9]. The WHO reports that millions of people around the world became infected with COVID-19 and millions of patients had died by December 2021 (https://who.sprinklr.com/, accessed on 27 December 2021).

The Russian Government assigned COVID-19 to the group of highly infectious diseases (Decree No. 66 of 31 January 2020) of the II group of pathogenicity, including such infections as plague, cholera and smallpox.

Suggestively, that patients with severe immune dysfunction caused by comorbid diseases are more likely to experience severe symptoms of COVID-19 than people with an adequate immune response, as it inhibits viral development at early stages [10]. At the same time, most of the research focuses on the risk of developing severe symptoms among persons with concomitant pathology (such as diabetes, cardiovascular diseases, chronic bronchopulmonary diseases and cancer) [11]. Some researchers point out that tuberculosis prevalence and mortality rates can increase during the COVID-19 pandemic [2], which is confirmed by WHO data in October 2021; they estimated that—because of the COVID-19 pandemic—global deaths from tuberculosis had increased for the first time in a decade [4]. However, tuberculosis incidence has actually decreased [11,12]. 

When constructing and analyzing mathematical models of COVID-19 and tuberculosis coinfection, it was shown that the most economically and clinically appropriate strategy is focusing on the prevention, treatment and control of combined infection. It is necessary to expand this strategy by including vaccination against COVID-19 and tuberculosis, taking into account the influx of infected immigrants, re-infection with exogenous tuberculosis and re-infection with COVID-19 after recovery [13].

## 2. Tuberculosis Diagnosis in the COVID-19 Pandemic

China, for example, is a country with a high burden of tuberculosis. However, its incidence from January to May decreased from 24% in 2019 to 13% in 2020 [14]. This may be related to a decrease in the number of people tested for tuberculosis and thus may not reflect the actual epidemic situation.

In fact, in a recent study by from McQuaid et al., the authors concluded that the reduction in tuberculosis deaths due to social distancing during this pandemic will not be as great as the damage caused by disruptions in healthcare delivery, and therefore, it is crucial to maintain and strengthen them during and after the COVID-19 pandemic [15]. 

Furthermore, tuberculosis hospitals were temporarily converted to COVID-19 diagnosis and treatment facilities which discouraged patients with suspected and confirmed tuberculosis from going to the hospital for fear of contracting COVID-19 [13]. The COVID-19 pandemic response, particularly containment measures, reassignment of health care personnel and equipment, is affecting tuberculosis prevention and care programs [16].

Similar symptoms in tuberculosis and COVID-19 (cough, hyperthermia, shortness of breath and chest pain) [1] might require additional examination and cause difficulties in differential diagnosis of two concomitant infections [16]. One study revealed that the most frequent complaints of patients with COVID-19 and tuberculosis were fever, headache, chest pain and cough [13]. Therefore, with an increase in the number of cases with both of these infectious diseases, failure to detect tuberculosis is possible during COVID-19 pandemic. In low- and middle-income countries where the burden of tuberculosis is highest, the differential diagnosis of COVID-19 and tuberculosis is key to detecting the co-infection and preventing misdiagnosis or even death [16]. In some publications, data about patients admitted to hospitals is presented with characteristics of acute COVID-19 symptoms [12]. 

In the study from China, which is one of the countries with a high tuberculosis burden, analyzed the severity of COVID-19 symptoms in individuals with latent tuberculosis infection (LTBI) and in patients with tuberculosis. Researchers indicated that severe symptoms of COVID-19 are more frequent in persons with LTBI (36% of cases) than in patients with diabetes (25%), hypertension (22%), coronary heart disease (8%) and chronic obstructive pulmonary disease (5%). Severe symptoms of COVID-19 in patients with tuberculosis was noted in 78% of cases [17]. It may be associated with the fact that LTBI+ individuals seropositive for SARS-CoV-2 infection exhibit heightened levels of humoral, cytokine and acute phase responses compared to LTBI- individuals. Thus, LTBI is associated with the modulation of antibody and cytokine responses as well as systemic inflammation in individuals seropositive for SARS-CoV-2 infection [18].

It is likely that the number of cases with combined bacterial and viral infections may increase during the COVID-19 pandemic, especially in countries with a high burden of tuberculosis infection [19], which was shown via modeling. Research indicates that COVID-19 may be combined not only with tuberculosis, but also with HIV, which can lead to a change in symptoms and radiological data. As a result, this will lead to significant changes in epidemiological indicators, which has been observed in the case of the combination of tuberculosis and HIV. This will require more active measures for the early detection of tuberculosis and viral infections, taking into account the development of secondary immunosuppression and damage to the lung tissue during COVID-19 infection. Over the next few years, we must promote bi-directional screening, multi-pathogen tests and protocols using combined solutions for tuberculosis /COVID-19 surveillance [20].

## 3. Pathogenesis of *M. tuberculosis (Mtb)* and SARS-CoV-2 Infections

Generally, Mtb affects the lung tissues with the proceeding creation of necrotic granulomas [21]. The granulomas are the focus of the anti- tuberculosis immune response and are presented by macrophages, neutrophils, dendritic cells (DCs), natural killer cells (NK), mast cells and lymphocytes. The correct treatment and effective immune response contribute to bacteria elimination and granuloma resolution; however, some patients’ characteristics can predispose them to the chronic processes. For example, neutrophils, due to ligand 1 of cell death (programmed death ligand 1 or PD-L1) expression, induce the loss of function and finally the death of lymphocytes [22]. In some patients with active tuberculosis, CD4+ percentage and absolute value reduction in the peripheral blood was found, which was related with the severity of infection [23].

Another mechanism of immune suppression was considered to be based on T regulators’ (Treg) role, which were shown to contribute to the infection development and persistence [24]. Anti-inflammatory processes were shown to be induced by CD8+ lymphocytes, which secreted IL-10 and TGF-β; moreover, ESAT-6 Mtb proteins cause the transformation of macrophages from phenotype M1, which produces IL-6, IL-12 and TNF-α, to M2, which is also capable of stimulating production of IL-10 [25,26].

The significant influence of the SARS-CoV-2 virus on the immune system resulting in severe immunosuppression, as well as the activation and progression of existing tuberculosis foci, can modify tuberculosis infection due to changes in the nature and intensity of the local cellular immune response such as in HIV infection in the AIDS stage when the lymphocytic, epithelioid cell and giant cell reactions become less intensive and alterative mechanisms of inflammation and rapid tuberculosis dissemination predominate [27].

Currently, it is known that the SARS-CoV-2 virus penetrates through the mucous membrane of the upper respiratory tract and replicates in the ciliated epithelium cells, with the further development of secondary viremia, immune disorders, hypoxia and dissemination to the target organs (heart, liver, kidneys, etc.) that leads to microangiopathy in the form of productive thrombovasculitis and hypercoagulation syndrome with immune system damage [10].

COVID-19 had a moderate impact on the clinical symptoms and prognosis of anti- tuberculosis treatment in the short term [1]. Detection of tuberculosis/COVID-19 coinfection requires the development of treatment algorithms to improve prognosis. It is important to take into account that the prolonged use of corticosteroids for the treatment of post-COVID-19 organizing pneumonia can lead to the reactivation of tuberculosis. In addition, doses of anti-tuberculosis drugs with hepatotoxic or nephrotoxic side effects should be adjusted in patients with severe COVID-19 with changes in liver and kidney function. In addition, lung fibrosis after COVID-19 can reduce the penetration of anti-tuberculosis drugs into the lungs and lead to adverse outcomes, especially in patients with MDR-TB [28].

During coronavirus infection, both direct damage to the cells of the respiratory system and that mediated by impaired blood circulation occurs. The direct cytotoxic effect of the virus is due to the penetration of the virus into cells expressing angiotensin-converting enzyme 2 (ACE2)-alveocytes, which leads to the development of pneumonia [11,27]. Unrestricted inflammatory infiltration of immune cells is observed in the lungs, which, in addition to direct viral damage, contributes to greater tissue damage due to the excessive secretion of proteases and reactive oxygen species. Diffuse alveolar damage is observed, characterized by the desquamation of alveolar cells, the formation of hyaline membranes and the development of pulmonary edema [29,30]. Extensive disturbance of microcirculation due to vascular damage and increased thrombus formation aggravates the damage to the lung tissue and reduces the effectiveness of the reparative processes [31,32]. 

In patients with severe symptoms of COVID-19, in contrast to patients with mild symptoms of the disease, in the vast majority of cases, the development of lymphopenia is characteristic, associated with a sharp decrease in the absolute content of CD4+ T cells, CD8+ T cells, B cells and natural killer cells [10,28].

It should be emphasized that the decrease in total levels of T cells, CD3+CD4+ and CD3+CD8+ cells in COVID-19 patients was especially significant in the elderly and in patients requiring intensive care [29]. Moreover, the CD3+ count was inversely related to the concentrations of the proinflammatory cytokines IL-6, IL-10 and TNFa in the blood serum, while the decrease in the level of these cytokines in the blood during recovery was closely associated with the restoration circulating T cell pool [33], which can be used as an important predictive criterion for the disease course [34].

It should be noted that neutrophils are the most important source of proinflammatory cytokines not only during infection with SARS-CoV-2, but also with Mycobacterium tuberculosis [35]. In the latter case, these cells are actively involved not only in the elimination of the pathogen at the initial stages of its invasion, but also in damage to the body’s own cells, leading to the destruction of lung tissue and its fibrosis [36]. Tuberculosis is one of the infectious diseases that primarily affects the lungs when the adaptive immune response carried out mainly by T cells is impaired [35]. Type 1 T-helpers (Th1) activate the production of IFNγ and TNFa, which contribute to the activation of antimicrobial protection by increasing the pool of connective tissue macrophages, while the activation of T-helpers 17 and peripheral blood neutrophils promotes the invasion of mycobacteria into the focus and, in addition to protective effects, damages surrounding tissues [28]. 

Changes in CD4+ T cell subsets during an infectious process are closely related to the effectiveness of the effector phase of the immune response against a particular type of pathogen [1,37]. Both with Mtb infection and the SARS-CoV-2 virus, Th1 cells are considered to be the key effector T-helper cells. Upon recognition of the specific antigen in peripheral tissues, they can produce IFNγ, which activates a very wide range of immunocompetent cells, including CD8+ cytotoxic T cells, ILC1, macrophages and B-cells involved in the elimination of intracellular pathogens. However, the role of this Th cell population in the pathogenesis of these diseases is rather controversial. In particular, among patients with tuberculosis, the relative number of Th2 cells in the peripheral blood was significantly increased, while the level of Th17 cells was significantly decreased, whereas no significant differences between the groups were observed in the case of Th1 and Tfh cells [38]. Similar results were obtained in the analysis of peripheral Th subsets in tuberculosis using nonspecific stimulation methods in vitro, which demonstrated decreased levels of CD4+IL-17A+ cells against the background of infection, while the content of CD4+IL-4+ lymphocytes in patients was significantly elevated [39]. However, there are also quite opposite observations: Wang T. et al. (2011) found an increased level of IL-17+CD4+ cells among tuberculosis patients compared with controls [40], which confirmed previously published studies on elevated IL-17 mRNA concentrations in peripheral blood lymphocytes among patients with active tuberculosis [41]. On the other hand, a decrease in the content of Th17 in the peripheral blood of patients was noted by two independent groups of researchers [42,43]. Furthermore, a decrease in the level of IL-17 in the peripheral blood of tuberculosis patients was closely associated with low treatment efficacy and unfavorable outcomes of this disease [44].

Among patients with COVID-19, an increase in the proportion of Th2 cells in circulation was associated with a poor prognosis, and patients who died had higher levels of CXCR3–CCR6– Th in circulation, while no significant differences were found in the percentage of Th1 or Th17 between the COVID-19 group and control patients [45]. 

It was also observed that, in patients with severe COVID-19, CD4+ Th cells were skewed toward CCR4-expressing Th2-like phenotypes within CD45RA+CD62L− and CD45RA–CD62L− cells, while central memory CCR6+ Th17-like cells were decreased when compared with healthy controls, while patients with moderate COVID-19 had no differences with controls [46]. 

Furthermore, a Th2 predominance and an increased blood level of Th2 may be closely associated with COVID-19 symptoms, such as intestinal hyperperistalsis, increased acidity of gastric juice and shortness of breath [47]. In addition, an over-reactive CXCR3–CCR6– Th2 cell response was an independent risk factor for death [48]. Moreover, COVID-19 convalescents continued to have elevated blood levels of Th2 cells for several months [43]. Another study demonstrated that CD4+ T cells from SARS-COV-2-infected patients accumulated IL-17A more efficiently in response to in vitro stimulation, as compared with cells of the respective subset in the comparison group [49]. At the same time, this publication also reported a decrease in the proportion of T-helpers carrying the key Th17 cell-surface markers, CD161 and CCR6, while the content of cells expressing Th2 markers (CCR4 and GATA3) was significantly higher than in the control group. Similar results were obtained using molecular biology methods; in particular, it was shown that the expression of Th17-associated genes decreased in peripheral blood CD4+ T cells of patients with severe COVID-19, as exemplified by RORC, IL17A, IL17F and CCR6 [50]. However, there is also contrary evidence of an increase in the proportion of Th17 and follicular T cells along with a slight decrease in the Th1 content in the blood of patients with COVID-19, while their Th2 values did not differ from those in the control group [51].

Currently, CCR6+ Th17 lymphocytes can be divided into four independent cell subsets, which differ in the levels of expression of the chemokine receptors CCR4 and CXCR3 and have different functional properties [52]. In particular, there are “classical” CCR4+CXCR3– Th17, “double-positive” CCR4+CXCR3 + or CCR6+DP Th17, “non-classical” CCR4–CXCR3+ or Th17.1, and “double-negative” CCR4–CXCR3– or CCR6+DN Th17 cells. We observed a decrease in the circulation of Th17.1 cells in patients with tuberculosis, while the levels of “classical” and CCR6+DP Th17 in the peripheral blood of patients significantly increased compared with the control group [38]. IFNγ-secreting Th17 lymphocytes were shown to negatively affect the development of long-term immune protection against repeated invasions of Mtb in animal experiments [53]. Of particular interest are the literature data that more than half of IFNγ-secreting Mtb-specific Th cells have the CXCR3+CCR6+ phenotype [28], which once again indicates the functional importance of Th17.1 cells in the development of the immune response in tuberculosis. It has been hypothesized that a shift in the Th1 to Th17.1 ratio towards a higher content of Th1 cells as compared with Th17.1 may contribute to the development of an effective immune response to the invasion of Mtb [54]. An analysis of circulating Th17 cell subsets in COVID-19 revealed a decrease in the proportion of Th17.1 cells, which are capable of producing IFNγ [55], although these results are not confirmed by publications of other authors, who found no significant differences in the content of these cells relative to the control group but did report an accumulation of “atypical” Th1 cells that had surface markers more characteristic of Th17 (or “non-classical” Th17.1 cells), such as CD161 and IL-1RI, in the peripheral blood of severely ill patients with pneumonia [51].Our analysis had shown that the relative number of CCR6+ Th17 cells was decreased within central and effector memory Th cells only in blood samples from patients with severe COVID-19 [46]. Furthermore, “classical” Th17 cells represented the most predominant Th17 subset in patients with severe COVID-19, whereas the levels of Th17.1 cells were dramatically decreased.

Multicolor flow cytometry allows to distinguish four subsets of cells differing in their phenotypic functional properties in the total Tfh pool: CXCR3+CCR6− Tfh1-, CXCR3−CCR6− Tfh2-, CXCR3−CCR6+ Tfh17-like and CXCR3+CCR6+ DP Tfh cells [56]. Disturbances in the functional activity of circulating Tfh cells were reported in a study by Kumar et al. (2014), who examined peripheral blood samples of patients with active tuberculosis [57]. In this study, not only the level of IL-21-producing cells in the blood of patients was reduced relative to the control values, but also the concentrations of this cytokine in the systemic circulation in the case of infection with Mtb were lower than those in the reference group. Furthermore, the level of IFNγ, whose production is usually associated with Tfh1 cells, was also decreased in patients [58]. An abnormal composition of Tfh subsets was also observed in patients with COVID-19, who had an increase in the proportion of CXCR3+CCR6– Tfh1 and CXCR3–CCR6– Tfh2 and a decrease in the level of CXCR3–CCR6+ Tfh17 relative to control group values [59]. Moreover, the frequencies of circulating Tfh cells were decreased within the central memory Th cells in patients with moderate and severe COVID-19, while Tfh17-like cells represented the most predominant Tfh-like subset in patients with severe COVID-19 [46]. These findings can probably explain the low effectiveness of Tfh cells in stimulating the humoral response, which is also associated with impaired formation of germinal centers in B-dependent areas of lymph nodes, as well as with a decrease in the expression of the key transcription factor Bcl-6, which is responsible for the implementation of the functional activity of Tfh [49]. In addition, the results of histological studies indicate atrophy of the germinal centers of the B-dependent zones in the lymph nodes in acute disease. It is also important to emphasize that the overwhelming majority of SARS-COV-2-specific Tfh cells belonged to CCR6+CXCR3– Tfh17 cells capable of stimulating the production of virus-specific IgA antibodies, but some of these cells had the Tfh1 phenotype (CCR6–CXCR3+), which had regulatory properties and the ability to suppress the immune response, as observed by Morito et al. [56]. Convalescent patients showing a high neutralizing ability were found to have a high number of c Tfh1 and c Tfh2 cells, whose high levels positively correlated with serum neutralizing activity [60]. We should mention that the dominance of circulating Tfh2 and/or Tfh17 subsets over Tfh1 was found in peripheral blood sample from patients with different systemic and organ-specific autoimmune diseases, including relapsing-remitting and secondary progressive multiply sclerosis [61], IgG4-related disease [62], pulmonary sarcoidosis [63], immunoglobulin A vasculitis [64] and psoriasis vulgaris [65]. These observations support not only the close link between altered Tfh subset balance and pathogenesis of human autoimmune diseases, but also point to the prolonged Tfh subsets alterations as a risk factor for post-COVID-19 symptoms and diseases associated with autoimmunity [66,67,68]. 

In tuberculosis, the adaptive immunity played the main role in the immune response, carried out mainly by T-lymphocytes [34]. With coronavirus infection, there is an increased depletion of T cells and a decrease in functional diversity [69]. According to various studies, viruses have been found in T-lymphocytes, macrophages and dendritic cells, which may also impair their function [70]. Therefore, coronavirus infection, against which cellular immunity is activated, leads to the depletion of the system aimed at combating tuberculosis (Figure 1).

In this case, immunosuppressive therapy of COVID-19 can accelerate the tuberculosis infection, which was shown in the description of clinical case of treatment with corticosteroids [71]. However, the study showed the opposite results [72]. Patients with tuberculosis and COVID-19 comorbidity were exposed to tuberculosis and SARS-CoV-2 proteins injection, and the level of INFγ was measured. tuberculosis and COVID-19 coinfection were shown to limit the ability to in vitro respond to SARS-CoV-2, while the Mtb-specific response was not impaired. The authors explained the results obtained by the immune response focused on tuberculosis infection, which weakened the response to SARS-CoV-2 [72].

On the other side, many researchers have shown that patients with concomitant pathologies develop severe irreversible pulmonary fibrosis during viral pneumonia significantly more often [9].

A possible explanation for this phenomenon is a pronounced exudation and a significant increase in the content of these cellular elements in the composition of polymorphic cell infiltration of the lung parenchyma, which is a manifestation of cell-mediated immune damage during the progression of viral pneumonia caused by SARS-CoV-2. 

Obviously, as the pneumonia progresses and the manifestations of diffuse alveolar damage increase, there is an increase in the pathological cellular infiltration of the interalveolar septa, and some of the cellular elements are located in the lumens of the alveoli (Figure 2).

Excessive migration of cellular elements in such circumstances can be the basis for a decrease in their content in the peripheral blood, and later on, with the progression of the pathological process, may lead to depletion of the organs of immunogenesis. The presence of an excessive number of immunocompetent cells in the lesion focus (lung parenchyma) expressing various biologically active substances, in turn, might be the basis for the preservation and deepening of diffuse alveolar damage. 

A pronounced macrophage reaction is following to the damage of the pulmonary parenchyma persists at all stages of viral pneumonia and is mainly represented by desquamated alveolar macrophages. However, as the diffuse alveolar damage deepens, which already started at the beginning of its late phase, the macrophage reaction of the epithelial lining of the alveoli is joined by cellular macrophage infiltration of the interstitium, apparently of monocytic origin (Figure 2I,J). Such a macrophage reaction, based on the likelihood of the development of typical pathological processes, is the basis for the onset of pathological fibrogenesis and the implementation of pneumofibrosis in the outcome of viral pneumonia [73]. The best method to study pathogenesis of tuberculosis -COVID 19 comorbidity is the creation of animal models to assess the impact of the SARS-CoV-2 on immune response during tuberculosis infection, the possibility of acceleration of tuberculosis infection. The most important aspects to be analyzed are the efficacy of diagnostic tools for the detection of tuberculosis infection based on immune response and treatment options, especially the use of immunosuppressive drugs. However, this area is not elucidated enough.

Pathak L et al. performed the study on dormant *M. tuberculosis* (dMTB) harboring mice with the murine hepatitis virus-1 (MHV-1) to evaluate the impact of viral infection on tuberculosis [74]. MHV-1 infection induced the reprogramming of lung mesenchymal stem cells (MSCs) with dMTB in stemness or altruistic stem cells (ASCs), which secreted the active bacteria. Thus, the authors assumed that SARS-CoV-2 infection might have the same effect and be the trigger of dormant tuberculosis.

## 4. Conclusions

A comprehensive analysis and modeling of the effects that the consequences of the COVID-19 pandemic have on the course of tuberculosis, HIV infection and malaria was carried out in one of the most significant publications [18], showing factors influencing the aggravation of the epidemic situation for all diseases being evaluated. 

In the current epidemic situation, the entire world community is facing the transmission of the new coronavirus infection (SARS-CoV-2), which has exacerbated the problems that existed before the pandemic [75,76,77]. Tuberculosis continues to threaten the lives and public health in many countries due to its characteristics being similar to those of COVID-19: airborne transmission, predominant lung damage, the development of secondary immunosuppression and infection dissemination. 

The question of the possible impact of vaccination against tuberculosis on the course and mortality from COVID-19 remains to be discovered. In fact, there is a version about the formation of systemic immunity in countries with an existing system of vaccination against tuberculosis. The system analysis showed that mortality from COVID-19 was lower in most countries with a BCG vaccination program that has been in place for the last 15 years. Mortality level from COVID-19 increased significantly in the population over 65 years of age. The researchers concluded that BCG vaccination has an impact on reducing COVID-19 mortality even for populations <40 years of age [78].

The real problems of tuberculosis in COVID-19 are:Fewer resources for the accurate diagnosis of new patients and management of already registered cases.Similar symptoms which complicate the diagnosisPathogenetic aspects of exacerbation of tuberculosis infection in COVID-19.Poor understanding of immune mechanisms and changes due to COVID-19 and tuberculosis co-infection presents challenges in infection control and treatment of these diseases.

Due to the need for the long-term isolation transmission of the COVID-19 pandemic, researchers stress paying special attention to establishing programs for diagnosis and treatment of tuberculosis in different countries where they have been reduced or completely stopped in the current epidemic situation [31]. 

One study showed a decrease in the incidence of tuberculosis, while another found patients with co-infection to have protective mechanisms against the development of a severe symptoms of COVID-19 but were immunocompromised to developing severe forms of tuberculosis, chronicity of the process and resistance to therapy [70,75]. Careful analysis of multiple publications cited in this study reveal the situation to be more complex. The COVID-19 pandemic has diverted resources away from the tuberculosis epidemic.

It is obvious that today we can be witnesses of changes in the established ideas regarding the processes of the prevention, diagnosis and prevention of the infectious diseases known to date that are familiar to the world community, which include tuberculosis. The whole world community is involved in a new hitherto unknown process of interaction with a new infectious agent—SARS-CoV-2. The spread of this virus may end with the emergence of the omicron strain, but the consequences of the COVID-19 pandemic in relation to other pathologies, including infectious ones, will be assessed for some time. Tuberculosis has historically been an infection that is sensitive to any changes in healthcare in any country. In countries where there is a problem of high prevalence of tuberculosis and HIV infection in the context of the COVID-19 pandemic, unforeseen features of the course of diseases may arise. These features may cause the reinfection of tuberculosis in people who have had tuberculosis earlier and in people with latent tuberculosis, especially in risk groups with comorbid pathology.

We recommend conducting further research to study changes in the clinical and morphological characteristics of tuberculosis among patients with a combination of two, and sometimes three, infectious diseases. Applying existing knowledge such as that gained through this study and expeditiously deploying new knowledge from future focused research is needed to prevent the worsening of the tuberculosis epidemiology during the COVID-19 pandemic. 

## Figures and Tables

**Figure 1 ijms-23-02235-f001:**
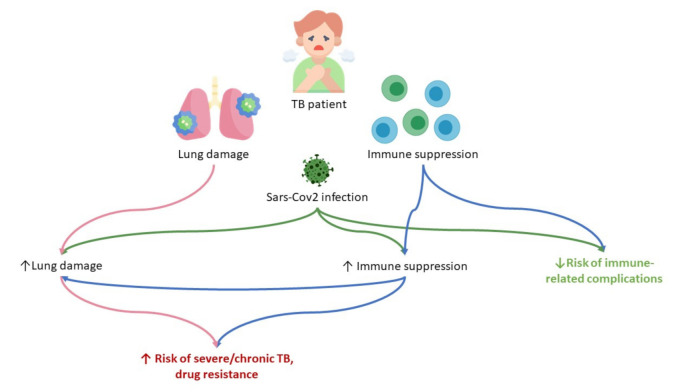
The supposed mechanisms of SARS-CoV-2 infection influence on tuberculosis infection. Tuberculosis infection contributes either lung damage and general immune suppression, which is enforced by SARS-CoV-2 infection and predisposes to the development of severe/chronic tuberculosis with drug resistance. On the other side, the latent immune suppression might be a defending factor against the development of cytokine storm.

**Figure 2 ijms-23-02235-f002:**
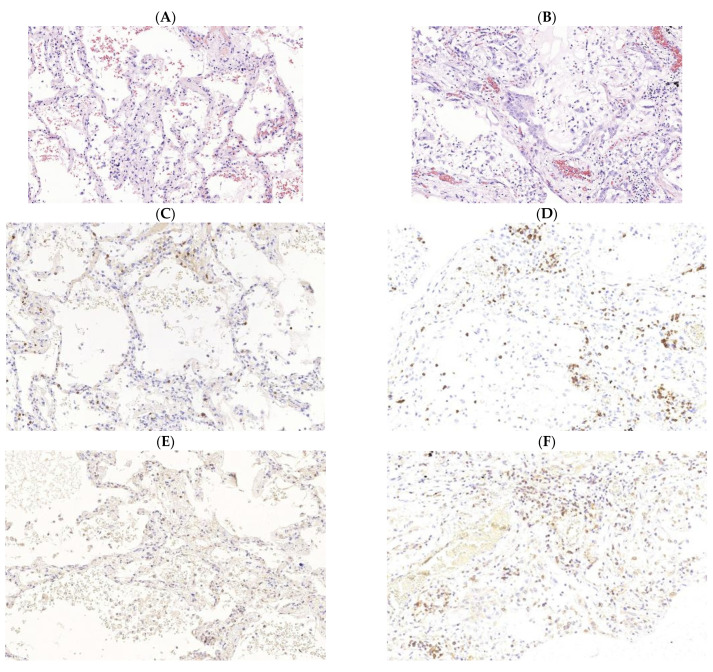
Immunomorphological characteristics of pathological cellular infiltration of the stroma of the lung’s respiratory parts among patients with new coronavirus infection caused by SARSCov2 (COVID-19). Acute (exudative) phase of diffuse alveolar damage (left). (**A**) staining with hematoxylin and eosin, ×400; (**C**) expression of CD3 (clone SP7, Diagnostic BioSystems, Pleasanton, CA, USA), ×400; (**E**) expression of CD4 (clone SP35, Cell Marque, Rocklin, CA, USA), ×400; (**G**) expression of CD8 (clone 144B, Diagnostic BioSystems, USA), ×400; (**I**) expression of CD68 (clone Kp-1, Cell Marque, USA. Late (proliferative) phase of diffuse alveolar damage (right). (**B**) staining with hematoxylin and eosin, ×400; (**D**) expression of CD3 (clone SP7, Diagnostic BioSystems, USA), ×400; (**F**) CD4 expression (clone SP35, Cell Marque, USA), ×400; (**H**) CD8 expression (clone 144B, Diagnostic BioSystems, USA), ×400; (**J**) CD68 expression (clone Kp-1, Cell Marque, USA), ×400.

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
