# Peer review of "Molecular and Cellular Mechanisms of M. tuberculosis and SARS-CoV-2 Infections—Unexpected Similarities of Pathogenesis and What to Expect from Co-Infection"

_ijms, 2022, doi:10.3390/ijms23042235_

Round 1
Reviewer 1 Report
Authors have focused much needed topic for the current situation. Authors focused on molecular and cellular differences and similarities of M.tuberculosis and SARS-CoV-2 infections. However, the article needs major revision.
- Authors could focus introduction more focus on interactions and research gap in the area of tuberculosis and SARS-CoV-2.
- TB prevalence rates should be updated to 2020.
- Authors focused the on more specifically on one country they should focus on global scenario and especially other TB endemic countries data as well.
- Lot of recent literature available on both molecular and immunological aspects of SARS-CoV-2 and TB coinfection.
- Pathogenesis section can be divided in to innate and adaptive responses.
- Authors could include proposed model like what are all the parameters involved in the pathogenies and impact of SARS-CoV-2 and TB on immune parameters.
- Also, authors could include impact of TB treatment on SARS-CoV-2 and TB coinfection.
- In the conclusion, authors should mention about the strength of their review article.
Author Response
1. Authors could focus introduction more focus on interactions and research gap in the area of tuberculosis and SARS-CoV-2
The answer: Some researchers point out that tuberculosis prevalence and mortality rates can increase during the COVID-19 pandemic (Glaziou, P. Predicted impact of the COVID-19 pandemic on global tuberculosis deaths in 2020. medRxiv and bioRxiv. DOI: 10.1101/2020.04.28.20079582), which is confirmed by WHO data - in October, 2021, they estimated that—because of the COVID-19 pandemic—global deaths from tuberculosis had increased for the first time in a decade (WHO. World Health Organization; Geneva: 2021. Global tuberculosis report 2021). The increase in deaths due to tuberculosis was mostly reported from 30 countries with the highest burden of the disease, including Angola, Congo, Bangladesh, China, and Brazil (Bagcchi S. Dismal global tuberculosis situation due to COVID-19. Lancet Infect Dis. 2021 Dec;21(12):1636. doi: 10.1016/S1473-3099(21)00713-1). However, tuberculosis incidence has actually decreased (Huang, C.; Wang, Y.; Li, X.; Ren, L.; Zhao, J.; Hu, Y.; Zhang, L.; Fan, G.; Xu, J.; Gu, X.; Cheng, Z.; Yu, T.; Xia, J.; Wei, Y.; Wu, W.; Xie, X.; Yin, W.; Li, H.; Liu, M.; Xiao, Y.; Gao, H.; Guo, L.; Xie, J.; Wang, G.; Jiang, R.; Gao, Z.; Jin, Q.; Wang, J.; Cao, B. Clinical features of patients infected with 2019 novel coronavirus in Wuhan, China. The Lancet. 2020, 395, 497-506. doi: 10.1016/S0140-6736(20)30183-5; Komiya, K.; Yamasue, M.; Takahashi, O.; Hiramatsu, K.; Kadota, J. The COVID-19 pandemic and the true incidence of tu-berculosis in Japan. Journal of Infection 2020, 81, e24-e25. doi: 10.1016/j.jinf.2020.07.004). When constructing and analyzing mathematical models of COVID-19 and tuberculosis coinfection, it was shown that the most economically and clinically appropriate strategy is focusing on the prevention, treatment and control of combined infection. It is necessary to expand this strategy by including vaccination against COVID-19 and tuberculosis, taking into account the influx of infected immigrants, re-infection with exogenous tuberculosis and re-infection with COVID-19 after recovery (Goudiaby MS, Gning LD, Diagne ML, Dia BM, Rwezaura H, Tchuenche JM. Optimal control analysis of a COVID-19 and tuberculosis co-dynamics model. Inform Med Unlocked. 2022;28:100849. doi: 10.1016/j.imu.2022.100849).
2. TB prevalence rates should be updated to 2020.
The answer: the publication presents the latest data on tuberculosis, which are presented in the WHO report for 2021 y. (reference 4, World Health Organization. Global tuberculosis report 2021. – Geneva: WHO (2021) 57. ISBN: 9789240037021).
3. Authors focused the on more specifically on one country they should focus on global scenario and especially other TB endemic countries data as well.
The answer: done.
4. Lot of recent literature available on both molecular and immunological aspects of SARS-CoV-2 and TB coinfection
The answer: The new data is added.
Generally, mbT affects the lung tissues with proceeding creation of necrotic granulomas (de Martino M, Lodi L, Galli L, Chiappini E. Immune Response to Mycobacterium tuberculosis: A Narrative Review. Front Pediatr. 2019;7:350. Published 2019 Aug 27. doi:10.3389/fped.2019.00350). The granulomas are the focus of the anti-TB immune response and are presented by macrophages, neutrophils, dendritic cells (DCs), natural kill-er cells (NK), mast cells, and lymphocytes. The correct treatment and effective immune response contribute to bacteria elimination and granuloma resolution, however, some patients’ characteristics can predispose to the chronic processes. For example, neutrophils due to ligand 1 of cell death (programmed death ligand 1 or PD-L1) expression induce the loss of function and finally the death of lymphocytes (Zhang Y, Zhou Y, Lou J, Li J, Bo L, Zhu K, et al. . PD-L1 blockade improves survival in experimental sepsis by inhibiting lymphocyte apoptosis and reversing monocyte dysfunction. Crit Care. (2010) 14:R220. 10.1186/cc9354). In some patients with active TB CD4+ percentage and absolute value reduction in the peripheral blood was found, which was related with the severity of infection (Venturini E, Lodi L, Francolino I, Ricci S, Chiappini E, de Martino M, et al. . CD3, CD4, CD8, CD19 and CD16/CD56 positive cells in tuberculosis infection and disease: peculiar features in children. Int J Immunopathol Pharmacol. (2019) 33:2058738419840241. 10.1177/2058738419840241).
Another mechanism of immune suppression was considered to be based on T regulators (Treg) role, which were shown to contribute the infection development and persistence (Parkash O, Agrawal S, Madhan Kumar M. T regulatory cells: Achilles' heel of Mycobacterium tuberculosis infection? Immunol Res. (2015) 62:386–98. 10.1007/s12026-015-8654-0). Anti-inflammatory processes were showen to be induced by CD8+ lymphocytes, which secreted IL-10 and TGF-β, moreover, ESAT-6 MbT proteins causes the transformation of macrophages from phenotype M1, which produces IL-6, IL-12 and TNF-α, into M2, which is also capable of stimulating production of IL-10 (Martinot AJ. Microbial offense vs host de-fense: who controls the TB granuloma? Vet Pathol. (2018) 55:14–26. 10.1177/0300985817705177; Refai A, Gritli S, Barbouche M-R, Essafi M. Mycobacterium tuberculosis virulent factor ESAT-6 drives macrophage differentiation toward the pro-inflammatory M1 phenotype and subsequently switches it to the anti-inflammatory M2 phenotype. Front Cell Infect Microbiol. (2018) 8:327. 10.3389/fcimb.2018.00327).
Therefore, coronavirus infection, against which cellular immunity is activated is assumed to lead to depletion of the system aimed at combating tuberculosis (Fig 1). In this case immune suppressive therapy of COVID-19 can accelerate the TB infection, which was shown in the description of clinical case of treatment with corticosteroids (Liu WD, Wang JT, Hung CC, Chang SC. Accelerated progression of pulmonary tuberculosis in a COVID-19 patient after corticosteroid treatment [published online ahead of print, 2021 Aug 26]. J Microbiol Immunol Infect. 2021;S1684-1182(21)00177-8. doi:10.1016/j.jmii.2021.08.007).
However, the study showed the opposite results (Petrone L, Petruccioli E, Vanini V, Cuzzi G, Gualano G, Vittozzi P, Nicastri E, Maffongelli G, Grifoni A, Sette A, Ippolito G, Migliori GB, Palmieri F, Goletti D. Coinfection of tuberculosis and COVID-19 limits the ability to in vitro respond to SARS-CoV-2. Int J Infect Dis. 2021 Dec;113 Suppl 1:S82-S87. doi: 10.1016/j.ijid.2021.02.090..). Patients with TB and COVID-19 comorbidity were exposed to TB and Sars-Cov2 proteins injection, and the level of INFγ was measured. TB and COVID-19 coinfection were shown to limit the ability to in vitro respond to SARS-CoV2, while M. tuberculosis-specific re-sponse wasn’t impaired. The authors explained the results obtained by the immune re-sponse focused on TB-infection, which weakened the response to SARS-CoV2 (Petrone L, Petruccioli E, Vanini V, Cuzzi G, Gualano G, Vittozzi P, Nicastri E, Maffongelli G, Grifoni A, Sette A, Ippolito G, Migliori GB, Palmieri F, Goletti D. Coinfection of tuberculosis and COVID-19 limits the ability to in vitro respond to SARS-CoV-2. Int J Infect Dis. 2021 Dec;113 Suppl 1:S82-S87. doi: 10.1016/j.ijid.2021.02.090..).
Therefore, an increase in the number of cases with both of these infectious diseases and overlooking of tuberculosis are possible during COVID-19 pandemic. In low and middle income countries where the burden of TB is highest, the differential diagnosis of COVID-19 and TB is key to detect the co-infection and prevent a bad evolution and even death (Cronin, A.M.; Railey, S.; Fortune, D.; Wegener, D.H.; Davis, J.B. Notes from the Field: Effects of the COVID-19 Response on Tuberculosis Prevention and Control Efforts — United States, March–April 2020. MMWR Morb Mortal Wkly Rep. 2020, 69, 971-972. doi: 10.15585/mmwr.mm6929a4., Koupaei M, Naimi A, Moafi N, Mohammadi P, Tabatabaei FS, Ghazizadeh S, Heidary M, Khoshnood S. Clinical Characteristics, Diagnosis, Treatment, and Mortality Rate of TB/COVID-19 Coinfectetd Patients: A Systematic Review. Front Med (Lausanne). 2021 Dec 1;8:740593. doi: 10.3389/fmed.2021.740593).
It may be associated with that LTBI+ individuals seropositive for SARS-CoV-2 infection exhibit heightened levels of humoral, cytokine and acute phase responses compared to LTBI- individuals. Thus, LTBI is associated with modulation of antibody and cytokine responses as well as systemic inflammation in individuals seropositive for SARS-CoV-2 infection (Chen, Y.; Wang, Y.; Fleming, J.; Yu, Y.; Gu, Y.; Liu, Ch.; Fan, L.; Wang, X.; Cheng, M.; Lijun Bi, L.; Liu, Y. Active or latent tu-berculosis increases susceptibility to COVID-19 and disease severity. MedRxiv.2020; https://doi.org/10.1101/2020.03.10.20033795., Rajamanickam A, Kumar NP, Padmapriyadarsini C, Nancy A, Selvaraj N, Karunanithi K, Munisankar S, Bm S, Renji RM, Ambu TC, Venkataramani V, Babu S. La-tent tuberculosis co-infection is associated with heightened levels of humoral, cytokine and acute phase responses in seropositive SARS-CoV-2 infection. J Infect. 2021 Sep;83(3):339-346. doi: 10.1016/j.jinf.2021.07.029. Epub 2021 Jul 28. PMID: 34329676; PMCID: PMC8316716).
5. Pathogenesis section can be divided in to innate and adaptive responses.
The answer: the authors considered the focusing on immune suppressive effects to be more informative to evaluate the impact of Sars-COv2 on TB infection without division in to innate and adaptive responses.
6. Authors could include proposed model like what are all the parameters involved in the pathogenies and impact of SARS-CoV-2 and TB on immune parameters.
The answer: the best method to study pathogenesis of TB-COVID 19 comorbidity is the creation of animal models to assess the impact of the Sar-Cov2 on immune response during TB infection, the possibility of acceleration of TB-infection. The most important aspects to be analyzed are the efficacy of diagnostic tools for the detection of TB-infection based on immune response and treatment options, especially the use of immune suppressive drugs. However, this area isn’t enough elucidated.
Pathak L et al performed the study on dormant Mycobacterium tuberculosis (dMTB) harboring mice with the murine hepatitis virus-1 (MHV-1) to evaluate the impact of viral infection on TB (Pathak L, Gayan S, Pal B, Talukdar J, Bhuyan S, Sandhya S, Yeger H, Baishya D, Das B. Coronavirus Activates an Altruistic Stem Cell-Mediated Defense Mechanism that Reactivates Dormant Tuberculosis: Implications in Coronavirus Disease 2019 Pandemic. Am J Pathol. 2021 Jul;191(7):1255-1268. doi: 10.1016/j.ajpath.2021.03.011). MHV-1 infection induced re-programming of lung mesenchymal stem cells (MSCs) with dMTB in stemness or altruistic stem cell (ASC), which secreted the active bacteria. Thus the authors assumed, that Sars-Cov2 infection might have the same effect and be the trigger of dormant tuberculosis.
7. Also, authors could include impact of TB treatment on SARS-CoV-2 and TB coinfection.
The answer: the information on medications for tuberculosis is not available presently, thus we could not address the effects of medications on the association between tuberculosis and COVID‐19 severity and mortality (Wang Y, Feng R, Xu J, Hou H, Feng H, Yang H. An updated meta-analysis on the association between tuberculosis and COVID-19 severity and mortality. J Med Virol. 2021 Oct;93(10):5682-5686. doi: 10.1002/jmv.27119)
COVID-19 had a moderate impact on the clinical symptoms and prognosis of anti-TB treatment in the short term (Stochino C, Villa S, Zucchi P, et al. Clinical characteristics of COVID-19 and active tuberculosis co-infection in an Italian reference hospital. Eur Respir J 2020: 2001708. doi:10.1183/13993003.01708-2020; Xu, Z.; Shi, L.; Wang, Y.; Zhang, J.; Huang, L.; Zhang, C.; Liu, S.; Zhao, P.; Liu, H.; Zhu, L.; Tai, Y.; Bai, C.; Gao, T.; Song, J.; Xia, P.; Dong, J.; Zhao, J.; Wang, F.S. Pathological findings of COVID-19 associated with acute respiratory distress syndrome. Lancet Respir Med. 2020, 8, 420-422. doi: 10.1016/S2213-2600(20)30076-X..). Detection of TB/COVID-19 coinfection requires the development of treatment algorithms to improve prognosis. It is important to take into account that prolonged use of corticosteroids for the treatment of post-COVID-19 organizing pneumonia can lead to reactivation of TB. In addition, doses of anti-tuberculosis drugs with hepatotoxic or nephrotoxic side effects should be adjusted in patients with severe COVID-19 with changes in liver and kidney function. In addition, lung fibrosis after COVID-19 can reduce the penetration of anti-tuberculosis drugs into the lungs and lead to adverse outcomes, especially in patients with MDR-TB. (Silva DR, Mello FCQ, D'Ambrosio L, Centis R, Dalcolmo MP, Migliori GB. Tuberculosis and COVID-19, the new cursed duet: what differs between Brazil and Europe? J Bras Pneumol. 2021 Apr 30;47(2):e20210044. doi: 10.36416/1806-3756/e20210044; Tamuzi JL, Ayele BT, Shumba CS, Adetokunboh OO, Uwimana-Nicol J, Haile ZT. Implications of COVID-19 in high burden countries for HIV/TB A systematic review of evidence. BMC Infect Dis. 2020;20(1):744–744. doi: 10.1186/s12879-020-05450-4).
8. In the conclusion, authors should mention about the strength of their review article.
The answer: it is obvious that today we can be witnesses of changes in the established ideas regarding the processes of prevention, diagnosis and prevention of the infectious diseases known to date, which include tuberculosis, that are familiar to the world community. The whole world community is involved in a new hitherto unknown process of interaction with a new infectious agent - SARS-CoV-2. The spread of this virus may end with the emergence of the omicron strain, but the consequences of the COVID-19 pandemic in relation to other pathologies, including infectious ones, will be assessed for some time. Tuberculosis has historically been an infection that is sensitive to any changes in healthcare in any country. In countries where there is a problem of high prevalence of tuberculosis and HIV infection in the context of the COVID-19 pandemic, unforeseen features of the course of diseases may arise. These features may cause reinfection of tuberculosis in people who have had tuberculosis earlier, in people with latent tuberculosis, especially in risk groups with comorbid pathology.
Reviewer 2 Report
Hello Dear
It is a good study and provides useful information. But it is better in relation to why tuberculosis is still prevalent in Russia after so many years. Tuberculosis testing, like AIDS, is one of the main tests in most office work in Russia, is it because there are so many old houses in Russia and the humidity is constantly felt everywhere? Or are there other reasons?
Is the a very high incidence of tuberculosis in China due to its high population?
And another question that could add to the appeal of this article is why leading countries in vaccine production, such as Russia or China, are relatively less receptive to vaccines.
And can tuberculosis be controlled by overcoming it? At what age?
Line 142: Why does this happen?
Het 200 needs further explanation.
The whole article is good and needs to add a few minor items, which of course will increase its scientific appeal.
Author Response
- But it is better in relation to why tuberculosis is still prevalent in Russia after so many years.
The answer to this question is rather complicated. Historically, in Russia the problem of tuberculosis is relevant permanently. In the 19th century there were no clear statistics and the figures were approximate. It is known that the death rate by the end of the 19th century was 250 per 100,000 population, which corresponded to figures in Europe. Thanks to the programs introduced in 1990 y., the incidence level was 34.2 and the death rate was 7.9 per 100,000 population. The well-known events of the “Perestroika” period led to an increase in these indicators in three times. (Yablonskii PK, Vizel AA, Galkin VB, Shulgina MV. Tuberculosis in Russia Its History and Its Status Today. Am J Respir Crit Care Med. 2015; 191(4):372–376. DOI: 10.1164/rccm.201305-0926OE). Significant measures and financial support from the Government of the Russian Federation have been taken to normalize the tuberculosis situation. In 2018 y., at the session of the United Nations Assembly in New York, Russia was recognized as a leader in the fight against tuberculosis (World Health Organization. Global tuberculosis report. - Geneva: World Health Organization. - 2019. - 283p. ISBN 978-92-4-156571 -4). The incidence level from 2011 y. to 2019 y. in the Russian Federation was not only stabilized, but decreased from 77.2 to 41.2 per 100 thousand population, respectively (Nechaeva OB The state and prospects of TB control service in Russia during the COVID-19 pandemic. Tuberculosis and Lung Diseases 2020;98(12):7-19 (In Russ.) https://doi.org/10.21292/2075-1230-2020-98-12-7-19). Obviously, the administrative resource, improving the quality of screening, diagnosis and treatment of tuberculosis makes a significant contribution to improving control over the spread of tuberculosis and reducing the incidence and mortality from the disease. Today, against the backdrop of restrictions associated with the detection of tuberculosis, there is a risk of deterioration in the epidemic indicators of tuberculosis not only in Russia, but also in other countries of the world. The presented data on the identification of immunological features of COVID-19 and tuberculosis can be significant for understanding the process of spread and course of tuberculosis infection in the future.
- Tuberculosis testing, like AIDS, is one of the main tests in most office work in Russia, is it because there are so many old houses in Russia and the humidity is constantly felt everywhere? Or are there other reasons?
The answer: historically, since the times of the former Soviet Union, in the Russian Federation the anti-tuberculosis system has been operating, the main work of which is to screen for tuberculosis infection, treat patients without bacteriological excretion and monitor people from risk group. In the other countries of the world, the same system does not exist, but exactly this structure in the country's healthcare system that can be a deterrent to a possible deterioration in the epidemiology situation of tuberculosis in Russia. It will be interesting to compare the effectiveness of this structure in the extreme conditions of the COVID-19 pandemic.
- Is there a very high incidence of tuberculosis in China due to its high population?
The answer: this fact may be due to the lack of the system similar to that in Russia for tuberculosis detection and treatment. China is more closed for control by WHO experts about TB. Perhaps the creation of a system for monitoring the diagnosis and treatment of tuberculosis infection in China can solve the issue of the spread of the disease in the future. The appearance of relevant data confirms this assumption (Jiang H, Zhang G, Yin J, Zhao D, Liu F, Yao Y, Cai C, Xu J, Li X, Xu W, Li W. Assessment of Strategies and Epidemiological Characteristics of Tuberculosis in Henan Province, China: Observational Study. JMIR Public Health Surveill 2021;7(1):e24830. doi: 10.2196/24830)
- And another question that could add to the appeal of this article is why leading countries in vaccine production, such as Russia or China, are relatively less receptive to vaccines.
And can tuberculosis be controlled by overcoming it? At what age?
The answer: in fact, there is a version about the formation of systemic immunity in countries with an existing system of vaccination against tuberculosis. The system analysis showed that mortality from the COVID-19 was lower in most countries with a BCG vaccination program has been in place last 15 years. Mortality level from the COVID-19 increased significantly in the population over 65 years of age. The researchers concluded that BCG vaccination has an impact on reducing the COVID-19 mortality even for populations <40 years of age. (Brooks NA, Puri A,Garg S, Nag S, Corbo J,Turabi AE, Kaka N, Zemmel RW, Hegarty PK, Kamat AM. The association of Coronavirus Disease‑19 mortality and prior bacille Calmette‑Guerin vaccination: a robust ecological analysis using unsupervised machine learning. Scientific Reports. 2021; 11(1): 774. DOI 10.1038/s41598-020-80787-z).
- Line 142: Why does this happen?
The answer: During coronavirus infection, both direct damage to the cells of the respiratory system and impaired blood circulation occurs. The lung damage is caused by the direct cyto-toxic effect of the virus, which penetrates into cells expressing ACE2 – alveocytes and hyperactivation of immune cells (Lin, L.; Lu, L.; Cao, W.; Li, T. Hypothesis for potential pathogenesis of SARS-CoV-2 infection - a review of immune changes in patients with viral pneumonia. Emerg Microbes Infect. 2020, 9, 727-732. doi: 10.1080/22221751.2020.1746199).
- Het 200 needs further explanation.
The answer: less IL17 is associated with poorer efficacy, which corresponds to what was written above about high increases in cells and cytokines during infection.